# The First Cold Atmospheric Plasma Phase I Clinical Trial for the Treatment of Advanced Solid Tumors: A Novel Treatment Arm for Cancer

**DOI:** 10.3390/cancers15143688

**Published:** 2023-07-20

**Authors:** Jerome Canady, Saravana R. K. Murthy, Taisen Zhuang, Steven Gitelis, Aviram Nissan, Lawan Ly, Olivia Z. Jones, Xiaoqian Cheng, Mohammad Adileh, Alan T. Blank, Matthew W. Colman, Keith Millikan, Cristina O’Donoghue, Kerstin M. Stenson, Karen Ohara, Gal Schtrechman, Michael Keidar, Giacomo Basadonna

**Affiliations:** 1Department of Translational Research, Jerome Canady Research Institute for Advanced Biological and Technological Sciences, Takoma Park, MD 20912, USA; drsmurthy@jcri-abts.com (S.R.K.M.); drtzhuang@jcri-abts.com (T.Z.); llawan@jcri-abts.com (L.L.); ozjones@jcri-abts.com (O.Z.J.); xcheng@usmedinnov.com (X.C.); 2Department of Mechanical and Aerospace Engineering, The George Washington University, Washington, DC 20052, USA; keidar@email.gwu.edu; 3Department of Surgery, University of Maryland, Capital Regional Medical Center, Largo, MD 21044, USA; 4Department of Surgery, Rush University Medical Center, Chicago, IL 60612, USA; steven_gitelis@rush.edu (S.G.); alan.blank@rushortho.com (A.T.B.); matthew_w_colman@rush.edu (M.W.C.); keith_millikan@rush.edu (K.M.); cristina_odonoghue@rush.edu (C.O.); kerstin_stenson@rush.edu (K.M.S.); karen_ohara@rush.edu (K.O.); 5Department of Surgical Oncology/General Surgery, Chaim Sheba Medical Center, Ramat Gan 52621, Israel; aviram.nissan@sheba.health.gov.il (A.N.); moh681@gmail.com (M.A.); gal.levi@sheba.health.gov.il (G.S.); 6Department of Surgery, University of Massachusetts Chan Medical School, Worcester, MA 01854, USA; giacomo.basadonna@umassmed.edu

**Keywords:** cancer treatment, surgical margin treatment, local regional recurrence, cold atmospheric plasma, solid tumors, breast cancer, ovarian, colon cancer, non-small cell carcinoma, sarcoma

## Abstract

**Simple Summary:**

It is estimated that 65% of solid tumor resections result in residual microscopic tumor cells at the surgical margin, which contributes to local recurrence and poor survival despite advancements in cancer therapies. Cold Atmospheric Plasma (CAP), a unique form of physical plasma, has emerged as a promising medical technology. Canady Helios Cold Plasma (CHCP) is a novel CAP device investigated in the first phase I clinical study with the primary goal of demonstrating safety. Promising findings demonstrated the device’s ability to control residual disease and improve patient survival. Ex vivo experiments on patient tissue samples showed CHCP-induced cancer cell death without harming normal cells. These results present CHCP as a safe and effective treatment in combination with surgery, providing a new avenue for controlling microscopic residual cancerous cells at the surgical margin.

**Abstract:**

Local regional recurrence (LRR) remains the primary cause of treatment failure in solid tumors despite advancements in cancer therapies. Canady Helios Cold Plasma (CHCP) is a novel Cold Atmospheric Plasma device that generates an Electromagnetic Field and Reactive Oxygen and Nitrogen Species to induce cancer cell death. In the first FDA-approved Phase I trial (March 2020–April 2021), 20 patients with stage IV or recurrent solid tumors underwent surgical resection combined with intra-operative CHCP treatment. Safety was the primary endpoint; secondary endpoints were non-LRR, survival, cancer cell death, and the preservation of surrounding healthy tissue. CHCP did not impact intraoperative physiological data (*p* > 0.05) or cause any related adverse events. Overall response rates at 26 months for R0 and R0 with microscopic positive margin (R0-MPM) patients were 69% (95% CI, 19–40%) and 100% (95% CI, 100–100.0%), respectively. Survival rates for R0 (n = 7), R0-MPM (n = 5), R1 (n = 6), and R2 (n = 2) patients at 28 months were 86%, 40%, 67%, and 0%, respectively. The cumulative overall survival rate was 24% at 31 months (n = 20, 95% CI, 5.3–100.0). CHCP treatment combined with surgery is safe, selective towards cancer, and demonstrates exceptional LRR control in R0 and R0-MPM patients. (Clinical Trials identifier: NCT04267575).

## 1. Introduction

Cancer remains a significant global health challenge, with over 1.9 million new cases, and is the second leading cause of death in the US, with 608,570 cancer-related fatalities in 2021 [1,2]. Malignant solid tumors are characterized by high local regional recurrence (LRR) and poor five-year survival rates [3]. Despite advancements in surgery, chemotherapy, radiation, immunotherapy, and molecular-targeted therapy regimes, residual disease can persist and lead to LRR. Today, the two fundamental challenges that remain after macroscopic tumor removal are the complete eradication of residual microscopic tumor cells and the preservation of noncancerous surrounding tissue.

Surgery remains a primary treatment for most solid tumors. However, complete removal (R0) of cancerous tissue is challenging, and residual microscopic tumors (R1) or macroscopic tumors (R2) at the surgical site can lead to LRR and poor prognosis. Solid tumor LRR for microscopic positive margins at the surgical site ranges from 5% to 65% [4]. Negative pathology findings do not guarantee the absence of LRR, which often requires re-operation, radiation, chemotherapy, or biological agents [5,6].

Plasma is the fourth state of matter formed by ionizing a neutral gas, i.e., (argon, helium) with electromagnetic fields [7]. High-temperature argon plasma coagulation has been used in endoscopy and surgery for over 40 years [8,9]. Cold atmosphere plasma (CAP), a non-equilibrium ionized gas composed of many different species, has made significant progress in clinical medicine since its introduction by physicists over 10 years ago [10,11]. Three-dimensional non-contact bioelectric pulse electromagnetic fields created by CAP cause a biophysical phenomenon, “Irreversible Electroporation (IRE)”, that increases cancer cell membrane permeability to CAP species such as reactive oxygen species (ROS) and reactive nitrogen species (RNS), resulting in apoptosis [12,13].

The efficacies of CAP devices for cancer treatment have been observed on many tumors, including sarcoma [14], glioblastoma [15,16], melanoma [17], breast [18,19], prostate [20], colorectal [21], lung [22], and many others [23,24,25,26,27]. The Canady Helios Cold Plasma (CHCP) system is a novel, non-thermal (26–30 °C), non-contact plasma jet [18] that generates a plasma-treated electromagnetic field (PTEF)™, ROS, RNS, and other species that eradicate microscopic tumor cells while sparing the surrounding noncancerous tissue by previously reported mechanism [28,29,30]. Standard CHCP treatment dosages for various cancer types have been developed in our laboratory based on previously published in vivo, in vitro, and in silico data that demonstrated CHCP-induced apoptosis reduced cell viability of cancer cells (80–90%) [14,18,19,23,30,31,32,33].

We report the first Phase I FDA-approved IDE clinical trial for the application of CAP in combination with surgical resection for the eradication of residual microscopic tumor cells.

## 2. Materials and Methods

### 2.1. Trial Oversight

The CHCP Trial was a multicenter, open-labeled, prospective controlled trial that enrolled eligible subjects with Stage IV or recurrent solid tumors. Patients were treated with CHCP intra-operatively at the surgical margin site after macroscopic tumor resection. The trial period was between March 2020–April 2021, with a follow-up ranging from 21–32 months (Figure 1). Twenty patients were recruited from Rush University Medical Center (RUMC), Chicago, IL, USA, and Sheba Medical Center (SMC), Tel HaShomer, Israel. Molecular biological studies were performed at the Jerome Canady Research Institute for Advanced Biological and Technological Sciences (JCRI-ABTS), Takoma Park, MD, USA. The trial was conducted in accordance with International Council for Harmonization Good Clinical Practice guidelines, and all patients provided written informed consent prior to enrollment. The authors vouch for the completeness and accuracy of the data and for the fidelity of the study to the protocol.

### 2.2. Patients

Inclusion criteria: Stage IV or recurrent solid tumors, male and female patients ≥18 years, biopsy verified histopathology or cytology diagnosis of a malignant solid tumor as defined by the World Health Organization (WHO) or by cross-sectional imaging reviewed by a board-certified radiologist, good performance status (Eastern Cooperative Oncology Group (ECOG) ≤ 2, Karnofsky > 60% and American Society of Anesthesiology (ASA)) score of ≤3, and scheduled for complete surgical resection. Additional eligibility criteria can be found at ClinicalTrials.gov identifier: NCT04267575.

Prior treatment evaluation: History and physical examination, baseline review, Chest X-ray, MRI or CT of the tumor site, and Chest and Abdominal CT and PET scan when appropriate. Surgical respectability and postoperative chemotherapy, radiation, or immunotherapy were determined between the surgeon and multidisciplinary management team.

### 2.3. Safety Assessment

The primary endpoint was safety. Intra-operatively, continuous monitoring of blood pressure (BP), pulse, body temperature, End Tidal CO_2_, and O_2_ saturation was performed while also recording the CHCP beam temperature using a Forward Looking InfraRed (FLIR) thermal camera (Appendix A). Biopsies of CHCP-treated margins of normal tissues were sent to RUMC or SMC Pathology Department for the detection of tissue damage. Patients were evaluated post-operatively up to 30 days from discharge according to Common Terminology Criteria for Adverse Events (CTCAE, version 4.03 to 5.0).

The secondary endpoint was sustained clinical complete response 12 months after CHCP combined with surgery. Pathological complete response was defined as the absence of residual cancer on the histologic examination of surgical specimens based on Response Evaluation Criteria in Solid Tumors (RECIST) [34]: Progressive disease, stable disease, partial response, near complete response, or complete response.

Overall Response Rate (ORR) is defined as no recurrence from the time of CHCP treatment to LRR in R0 resected patients. ORR was evaluated by postoperative T2 weighted MRI, PET, or CT scans. LRR-free survival includes events of LRR or death from the time of CHCP treatment. Overall Survival (OS) was defined as the duration from the intraoperative CHCP treatment to death.

### 2.4. Pathology

H&E staining: Fresh tissue specimens with/without ex vivo CHCP treatment were fixed in 10% neutral buffered formalin for 24–48 h, followed by dehydration in graded ethanol solutions and clearing with xylene and embedded in paraffin wax. Thin sections (6–7 μm thick) were obtained and underwent H&E staining involving deparaffinization, hematoxylin staining, differentiation, eosin staining, dehydration, and mounting. The stained slides were analyzed under a light microscope to evaluate cellular structures, tissue architecture, and pathological changes.

Terminal deoxynucleotidyl transferase dUTP nick-end labeling (TUNEL) Assay: Formalin-Fixed Paraffin-Embedded (FFPE) sections were deparaffinized using xylene and graded ethanol solutions and rehydration with distilled water. The TUNEL reaction mixture (Abcam, Cambridge, MA, USA), composed of the TUNEL enzyme and nucleotide mixture, was applied to the tissue sections, followed by incubation at 37 °C. Chromogenic development involved washing the sections and applying enzyme-conjugated substrate solution diaminobenzidine (DAB). Sections were counterstained with a methylene green, followed by dehydration, and mounted for analysis.

### 2.5. Primary Culture

Ex vivo CHCP-treated tumors samples were dissociated and processed using Miltenyi Biotec GentleMACS Dissociator (Miltenyi Biotech, Gaithersburg, MD, USA), and tumor cells were isolated using the Tumor Cell Isolation Kit (Miltenyi Biotech, NO. 130-108-339, Gaithersburg, MD, USA). The tissues were sliced into small fragments (2–4 mm^3^) in RPMI-1640 medium (Life Technologies, Grand Island, NY, USA) supplemented with Penicillin–Streptomycin (100 IU/mL) (Life Technologies, Grand Island, NY, USA) and Amphotericin B (0.25 ug/mL) (Life Technologies, Grand Island, NY, USA) followed by filtration and centrifugation. Cell pellets were plated on collagen-coated culture plates in DMEM-F12 (Life Technologies, Grand Island, NY, USA) or RPMI-1640 supplemented with 10% fetal bovine serum (Sigma 12306C, Saint Louis, MO, USA) and incubated at 37 °C with 5% CO_2_ for 3 to 16 days. The cells were imaged and counted for analysis. Cell count or growth area was plotted as Mean ± Standard Error of the Mean (SEM) (n = 3).

### 2.6. Quantitative Confocal Immunofluorescence Analysis

FFPE sections were deparaffinized and permeabilized with permeabilization buffer (0.2% Triton X-100, Milwaukee, WI, USA in PBS) and blocked with a blocking (5% bovine serum albumin (BSA) (Sigma 12306C, Saint Louis, MO, USA)). Fluorescently labeled primary antibodies (specific CD44 and BID) (Cell Signalling, Danvers, MA, USA) at dilution of 1:100 in dilution buffer were incubated with the samples overnight. After washing thoroughly, the samples were then mounted with DAPI mounting media (Vector Laboratories, Inc., Newark, CA, USA) The stained sections were imaged under confocal microscope, high-resolution images were acquired, and fluorescence intensity was quantified.

### 2.7. Statistical Analysis

Kaplan–Meier curves with 95% confidence intervals (CI) were generated with RStudio, Version 1.4.1106. Quantitative data were analyzed by Student’s *t*-test with differences considered statistically significant for * *p* < 0.05, ** *p* < 0.01, and *** *p* < 0.001.

## 3. Results

### 3.1. Patient Demographics and Characteristics

Five patients were treated at SMC, and 15 patients at RUMC. Patient details are shown in Table 1.

### 3.2. Safety Outcomes

#### 3.2.1. Intra-Operative Physiological Data and CHCP Temperature

BP, pulse, body temperature, End Tidal CO_2_, and O_2_ saturation were recorded before, during, and after CHCP treatment, which demonstrated no significant changes (Appendix A, *p* > 0.05).

#### 3.2.2. Post-Operative Adverse Events

There were no adverse events related to CHCP treatment. Two patients had surgical complications (Clavien Dindo Grade III and IIIb); refer to Table 1.

### 3.3. Efficacy Outcomes

#### 3.3.1. Recurrence and Overall Survival

Twelve patients underwent R0 resection. JCRI-ABTS reported 5/12 (42%) R0 patients with microscopic positive margins (R0-MPM). Two out of seven (29%) R0 patients exhibited LRR between 1–20 months, and no R0-MPM patients exhibited LRR to date. The 26-month ORR for R0 and R0-MPM patients was 69% (95% CI, 19–40%) and 100%, respectively (95% CI, 100–100.0%) (Figure 2A). Comparison of ORR and LRR-free survival between R0 patients and those with R0-MPM did not reveal any statistically significant differences, with median durations of 21 and 25 months, respectively (Figure 2A,B). As of 13 February 2023, 10/20 (50%) patients succumbed to their disease within a range of 3 to 32 months. The Kaplan–Meier analysis (Figure 2C) indicated a 31-month overall survival (OS) rate of 24% (95% CI, 5.3–100.0%) with a median survival of 23 months. Survival rates at 28 months varied among patient groups: R0 (86%), R0-MPM (40%), R1 (67%), and R2 (0%) (*p* < 0.013) (Figure 2C). Only R0 and R0-MPM patients showed significantly higher survival rates compared to R2 patients at 10 months (*p* < 0.0019 and *p* < 0.0083, respectively). No significant difference in survival rate was found among R1 and R2 patients at 10 months or among R0, R0-MPM, and R1 at 28 months. The median survival for R0, R0-MPM, R1, and R2 patients were 23, 25, 25.5, and 6.5 months, respectively.

#### 3.3.2. Histopathology Evaluation

RUMC and SMC Pathology Department reported no thermal damage or histological changes in 19/21 (90%) samples of ex vivo CHCP-treated normal tissue. Two samples demonstrated thermal damage, but a Bovie electrocautery device was used to dissect the tissue prior to CHCP treatment.

#### 3.3.3. Histopathology Findings

A total of 12 patients (60%) achieved R0 status. Tissue specimens were collected as illustrated in Figure 3A: Tumor (resected tumor tissue), Zone 0 (surgical margin), Zone 1 (1 cm away from the surgical margin), and Normal (surrounding normal tissue of the tumor). Resected tissues were CHCP treated ex vivo. The electroporation and cellular mechanism of CHCP in cancer cells is illustrated in Figure 3B,C and reported in our previous studies [30,33]. H&E stained tissue samples demonstrated CHCP-induced tumor cell death in 15/15 samples (100%) and histological damages on normal tissue in 0/15 samples (0%). Tumor cells were detected in the following areas: Zone 0 in 5/9 (55%), Zone 1 in 0/6 (0%), and normal tissue in patient R0010 (Tensor Fasciae Latae) 1/13 (7.7%). No thermal damage in normal tissue was found after CHCP treatment (Figure 3D). A complete overview of H&E-stained CHCP-treated and untreated tissues is available in Appendix A.

#### 3.3.4. Primary Tissue Culture

The primary culture was established successfully from 10/15 (67%) tumor samples (Figure 4A). Tumor cell survival from the CHCP-treated tissues was significantly lower compared to the untreated tumor tissues in 8/10 (80%) cases, no cancer cells survived in 3/10 (30%) cases, and significantly fewer cells survived in 5/10 (50%) cases (Appendix A). CHCP did not significantly reduce cancer cell survival in 2/10 cases (20%) due to suboptimal treatment doses.

#### 3.3.5. Apoptotic DNA Damage Analysis by TUNEL Assay

Ex vivo CHCP-treated samples were analyzed for apoptosis by TUNEL assay, and positive cells were measured (Figure 4). Less than 1% of the untreated and treated normal cells showed TUNEL staining (*p* > 0.05). CHCP-treated tumor tissue showed a significant increase in TUNEL-positive cells compared to untreated tumor tissue (54–88%), demonstrating CHCP-induced DNA damage and apoptosis in cancer cells (*p* < 0.001). A complete comparison table demonstrating all patient tissue samples with TUNEL staining of CHCP-treated and untreated tumor samples is available in Appendix A.

#### 3.3.6. Confocal Immunofluorescence Analysis

Quantitative confocal immunofluorescence analysis showed that tumor stem cell marker CD44 was significantly decreased, and the apoptotic marker BID protein expression was increased in CHCP-treated tumor samples compared to untreated tumor samples (Figure 5). Significant morphological alterations such as shrinkage or fragmentation in the nuclear morphology of cells in the CHCP-treated tumor tissues were observed, suggesting chromatin destabilization during apoptosis.

#### 3.3.7. Gene Expression Analysis

CHCP-treated tumor samples were analyzed for gene expression using real-time qPCR to evaluate 111 genes related to apoptosis, oxidative stress, immune checkpoint, T-cell regulation, and tumor promotion. Complete details of the study, including a table of differentially regulated genes, are provided in Appendix A.

#### 3.3.8. Regulation of Protein Expression

CHCP-treated and control tissue samples were analyzed using Western blot for the expression of apoptotic protein markers (BAD, BCL2, PUMA, and BID). Some cancer types exhibited significant up-regulation of pro-apoptotic proteins and down-regulation of the anti-apoptotic marker BCL2 following CHCP treatment. Appendix A provides detailed data and analysis.

## 4. Discussion

The LRR rate for solid tumors remains as high as 65%, even in the absence of distant disease [4]. Historically, stage IV solid tumors are typically not resected due to the high risk of palliative surgery [35]. We demonstrated that CHCP treatment in combination with surgery is a viable option for controlling LRR and possibly improving overall survival.

CHCP is a safe and potentially effective treatment option for a wide range of Stage IV or Recurrent solid tumors. Although R0 resection was achieved in 60% of patients (n = 12), microscopic tumor cells were identified at the surgical margin site (Zone 0) in 5 patients (R0-MPM, 42%) (R0004, R0005, R0009, R0011, and R0012). Despite the OS rate of 40% at 28 months observed in R0-MPM patients, none of the patients exhibited any signs of recurrence (LRR) during the follow-up period. The absence of LRR in these five R0-MPM patients following the CHCP therapeutic approach could completely eradicate undetectable microscopic tumor cells at the surgical margin site, independent of the cancer type.

Ex vivo observations of H&E and TUNEL staining showed significant cell death at Zone 0. Molecular and cell biology analyses demonstrated that CHCP treatment induced differential expression of pro-apoptotic and anti-apoptotic genes. These findings provide compelling evidence of CHCP-induced apoptosis in cancer cells, underscoring its potential to eliminate microscopic tumor cells at the surgical margin.

The 26-month non-LRR rate of R0 and R0-MPM patients reached 57% and 80%, respectively. There was no significant difference in OS rate between R0, R0-MPM, and R1 patients at 28 months. Furthermore, overwhelming significance in OS was observed when comparing R0, R0-MPM, R1, and R2 resection, with rates of 86%, 40%, 67%, and 0% at 28 months, respectively. These data underscore the heightened therapeutic efficacy of CHCP treatment when combined with R0, R0-MPM, and R1 surgical resection. Consequently, our study highlights the potential of expanding CHCP treatment to encompass less invasive solid tumors, as it offers promise for enhancing local control.

CHCP is a non-invasive and non-contact electroporation device that creates PTEF™, ROS, RNS, and other species which can be personalized to the patient’s cancer or subtype. CHCP treatment takes only 5–7 min intra-operatively and preserves the surrounding noncancerous tissue with no side effects in contrast to contact-based electroporation devices such as Optune (Novocure), NanoKnife (Angiodynamics), and Aliya System (Galvanize Therapeutics). CHCP seamlessly integrates into existing standard-of-care protocols without adding extra burden or disrupting the patient’s overall treatment regimen.

The JCRI-ABTS laboratory discovered two mechanisms underlying the effects of CHCP-generated electromagnetic fields on membrane permeability: electroporation and alteration of membrane lipid structure [33], consistent with previous studies on CAP [12,13]. We provide valuable insight into the biological effects of PTEF™ and the mechanisms underlying its ability to induce membrane permeability in living cells [36,37,38].

Furthermore, our laboratory discovered that CHCP-treated cancer cells undergo histone mRNA oxidation and degradation, leading to the upregulation of DNA damage response genes and apoptosis [30]. Compared to normal cells, cancer cells have frequent cell cycles and a greater percentage of cells in the S-phase, and higher levels of intracellular ROS and RNS [39]. Additional CAP-generated species may overwhelm the system and switch the ROS and RNS effect from tumor-promoting to tumor-suppressing [40].

Studies on CAP treatment of various cancer types have demonstrated the abscopal effect in tumors, suggesting its potential to induce an immune response by releasing cytokines and DAMPs from apoptotic cancer cells [41]. However, immunosuppressants may hinder this effect, possibly contributing to early LRR in patient R0013 with a history of immunosuppressant use.

We acknowledge the study limitation of 20 patients, and a phase II study will be conducted with a larger racially, ethnically diverse cohort of patients to determine CHCP efficacy on different cancer molecular subtypes.

## 5. Conclusions

This clinical trial opens an important window into the potential use of this technology following Stage IV solid tumor cytoreduction and primary tumorectomy. These results highlight the safety, selectivity, and significant reduction of local regional recurrence achieved through the synergistic approach of CHCP treatment and surgery. Our comprehensive findings underscore the remarkable capacity of CHCP to eradicate the possibility of recurrence, even when microscopic tumor cells are detected at the surgical margin site in R0 patients. The combination of CHCP potency on cancer cells and the benign interaction with normal tissue renders it an exceptionally enticing therapeutic approach for addressing surgical margins, thereby mitigating the risk of recurrence.

## Figures and Tables

**Figure 1 cancers-15-03688-f001:**
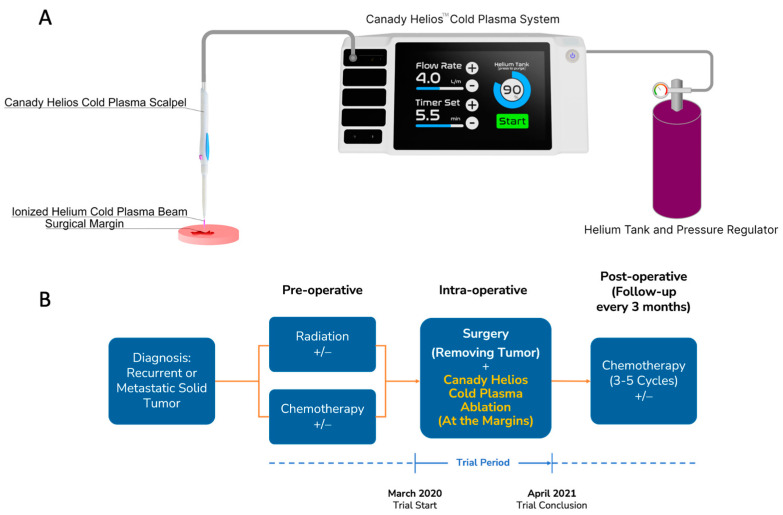
(**A**) Device setup of Canady Helios™ Cold Plasma System. (**B**) Study protocol showing pre-operative, intra-operative, and post-operative treatments. The patients included in the trial underwent multiple rounds of chemotherapy and radiation before being considered for Canady Helios Cold Plasma (CHCP) treatment. FDA approval was specifically granted for the use of CHCP at the surgical margins after macroscopic removal of solid tumors in patients with Stage 4 metastatic or recurrent solid tumors. Patient selection was based on a comprehensive evaluation conducted by a multi-disciplinary team at each institution. Eligible patients had a treatment history including chemotherapy, radiation, immunotherapy, surgery, and Hyperthermic Intraperitoneal Chemotherapy (HIPEC). Notably, these patients would not traditionally have been offered surgery due to the advanced stage of their disease. The primary objective of the CHCP trial was to demonstrate the safety of the treatment. Achieving a complete (R0) or microscopic residual (R1) surgical resection with CHCP treatment was to decrease the likelihood of local regional recurrence (LRR).

**Figure 2 cancers-15-03688-f002:**
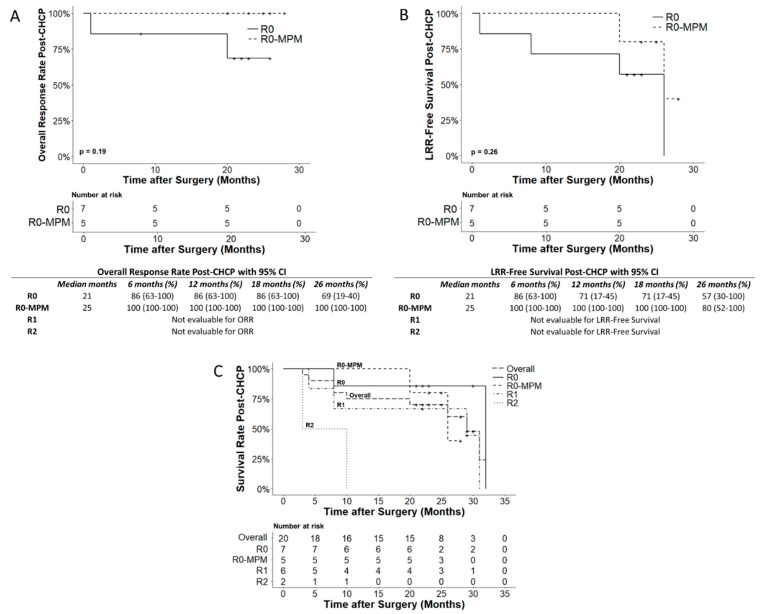
Kaplan–Meier survival curves as of February 13, 2023. (**A**–**C**) Symbol ‘+’ indicates the last follow-up of each individual patient. (**A**) ORR of patients with R0 (n = 7) and R0-MPM status (n = 5) (*p* > 0.05). Early recurrence (<20 months) occurred in two R0 patients: R0013 had a previous kidney transplant on immunosuppression, and R0002 with metastatic myxofibrosarcoma. (**B**) LRR-free survival rate (including events for recurrence or death) for patients with R0 (n = 7) and R0-MPM status (n = 5) (*p* > 0.05). (**C**) OS rates of all patients (n = 20) and among the subset of R0 (n = 7), R0-MPM (n = 5), R1 (n = 6), and R2 (n = 2) patients: R0 vs. R2 (*p* < 0.0019), and R0-MPM vs. R2 (*p* < 0.0083) showed significant difference at 10 months.

**Figure 3 cancers-15-03688-f003:**
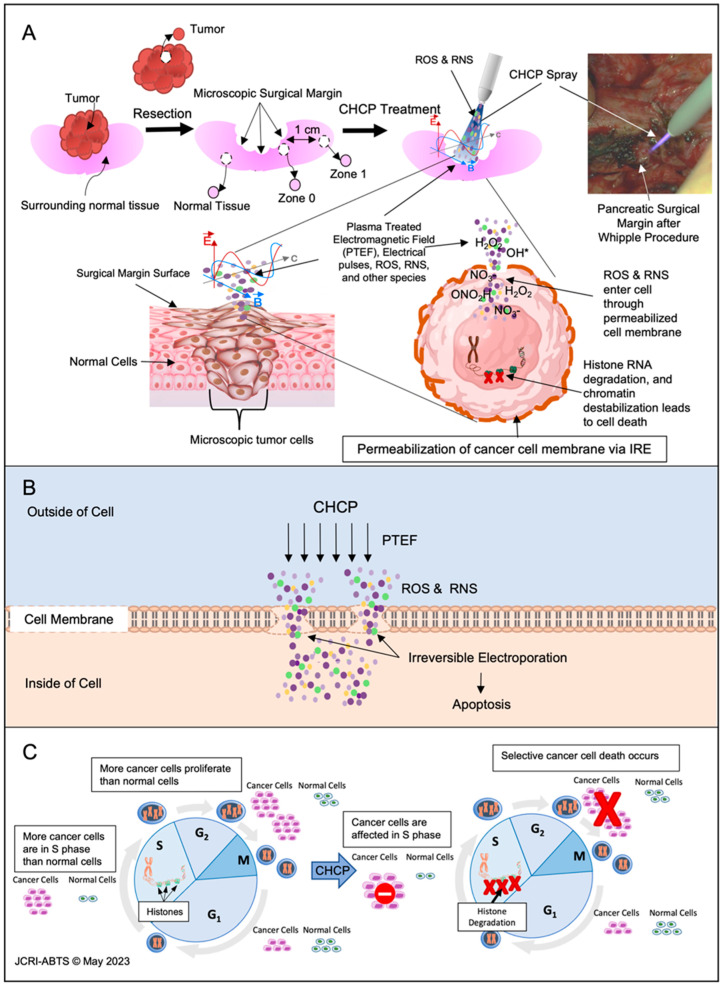
(**A**) Schematic images of the surgery and CHCP treatment procedure: resection of the tumor using standard surgical procedures, collection of specimens (Tumor, Zone 0, Zone 1, and Normal tissue), and application of CHCP spray to the surgical margin and a representative picture of CHCP treatment performed on pancreatic surgical margin after Whipple procedure for cancer. The OH* denotes free form of hydroxyl; (PTEF: E is the electrical field, B is the magnetic field, C is the 3D reference of the magnetic field.) (**B**) Illustration of CHCP produced PTEF™, ROS, and RNS entering a cancer cell membrane through “pores” generated by electroporation. (**C**) Illustration of the selective induction of cancer cell death by CHCP treatment resulting from 8-oxoG modification and degradation of histone mRNA during the early S phase. (**D**) Light micrographs displaying H&E staining of Patient R0004 (metastatic recurrent non-small cell lung adenocarcinoma to the left hip/upper end of femur) Tumor, Zone 0, Zone 1, and Normal (vastus lateralis muscle) tissue samples with or without CHCP treatment. Black arrows and yellow arrows point to untreated tumor cells and treated dead tumor cells, respectively (scale bars = 0.2 mm).

**Figure 4 cancers-15-03688-f004:**
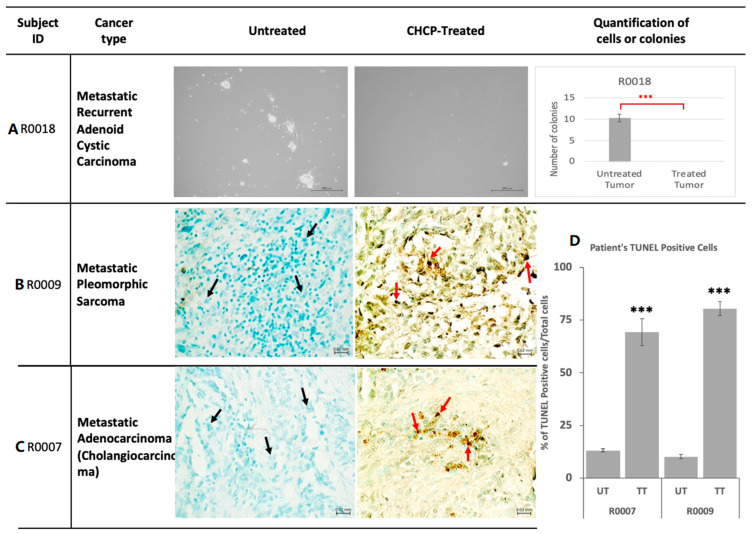
(**A**) Representative phase contrast images and quantification of primary culture established with Patient R0018 Tumor Samples, untreated and post-CHCP treatment (scale bars = 200 μm). (**B**,**C**) represents light microscope images of apoptosis analysis by TUNEL assay staining for patient R0009 (**B**) and R0007 (**C**) tumor tissue samples that were untreated (UT) or treated (TT). Black arrows and red arrows point to untreated tumor cells and treated dead tumor cells, respectively. Apoptotic cells were labeled with TdT, and signals were developed using DAB with Methylene Green counter stain. The majority of CHCP untreated tissue sections (0.1–24%) did not show TdT signal. Light Micrographs displaying the morphological Spectrum at 63×. (**D**) Quantitative analysis of apoptosis demonstrated by TUNEL assay. Percentage of TUNEL-positive cells (54–88%) signified DNA damage due to apoptosis after CHCP treatment. There was a significant increase in TUNEL-positive cells in TT compared to UT. Semi-quantitative analysis was performed on TUNEL-stained FFPE sections by manually counting cells under a light microscope. The data are represented by the SEM (n = 3) (*** *p* < 0.001, Student’s *t*-test).

**Figure 5 cancers-15-03688-f005:**
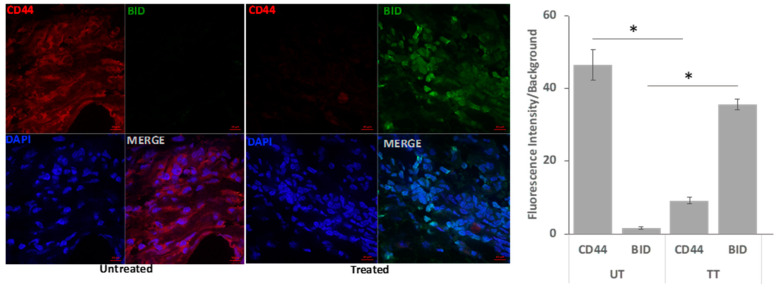
Expression of apoptotic marker protein BID After CHCP treatment in metastatic recurrent colon cancer patient tumor tissue: Representative multicolor fluorescence confocal images showing tumor cells stained for CD44 (red), BID (green), and DAPI (blue). The CHCP-treated tumor tissue shows increasing protein staining for apoptotic protein BID, and untreated samples show increased staining for tumor marker CD44. Statistical analysis by SEM (n = 12) (* CD44: UT vs. TT *p* < 0.00643^−6^, BID: UT vs. TT *p* < 0.00156^−12^ Student’s *t*-test).

**Table 1 cancers-15-03688-t001:** Table for patient demographics, characteristics, and surgery details.

Patient ID (Sex, Age)	Post-Operative Diagnosis	R0	Post Operation ECOG ≤ 2 ASA ≤ 3	LOS (Days)	OS Post Surgical Resection	Adverse Events with CHCP	Preoperative Staging Stage IV (Metastatic or Recurrent)	Intra- and Post-Operative Outcomes	Neo-Adjuvant Chemo-Therapy	Intra-Operative Chemo-Therapy	Adjuvant Chemo-Therapy
**S0001** **(F, 49)**	Metastatic recurrent colon cancer	Yes	Yes	7	Died of disease (32 months) Lost follow-up (26 months)	None	Yes	No complications	CT	HIPEC, IORT	CT
**S0002** **(F, 44)**	Metastatic recurrent ovarian cancer	No	Yes	5	Died of disease (29 months)	None	Yes	No complications	CT	None	CT
**S0003** **(F, 61)**	Metastatic anal cancer to the liver	No	Yes	5	Died of disease (3 months)	None	Yes	No complications	CT	None	IMT
**S0004** **(M, 51)**	Recurrent epithelioid peritoneal Mesothelioma	No	Yes	1	Died of disease (10 months)	None	Yes	No complications	CT	HIPEC	None
**S0005** **(M, 60)**	Recurrent Colon Cancer	No	Yes	6	Died of disease (31 months)	None	Yes	No complications	CT	HIPEC	CT
**R0002** **(F, 85)**	Metastatic myxofibrosarcoma left chest wall	Yes	Yes	1	Alive with disease (30 months) Recurrence of disease (20 months) at left chest wall resection site	None	Yes	No complications	RT, RT (internal), CT	None	RT
**R0003** **(F, 63)**	Recurrent metastatic breast carcinoma to the right pelvis	No	Yes	5	Alive with disease (29 months) FDRS^9^	None	Yes	No complications	RT, HT	None	RT, HT, IMT, CT
**R0004** **(F, 76)**	Metastatic non-small cell lung carcinoma (NSCLC) to the left proximal femur	Yes	Yes	4	Died of disease (26 months)	None	Yes	No complications	None	None	RT, IMT
**R0005** **(M, 61)**	Metastatic renal cell carcinoma to the left clavicle	Yes	Yes	2	Alive (28 months) FDRS	None	Yes	No complications	RT	None	IMT
**R0007** **(M, 59)**	Metastatic cholangiocarcinoma to the left distal humerus	No	Yes	3	Died of disease (8 months)	None	Yes	No complications	None	None	CT, CRT
**R0008** **(M, 58)**	Metastatic Non-small cell Lung Carcinoma (NSCLC) to the right hip and spine.	Yes	Yes	7	Died of disease (8 months)	None	Yes	No complications	CRT, IMT, RT, CT	None	CT
**R0009** **(F, 77)**	Pleomorphic sarcoma of left distal femur	Yes	Yes	2	Alive (25 months) FDRS	None	Yes	No complications	None	None	RT
**R0011** **(M, 67)**	Metastatic Chordoma to right gluteal	Yes	Yes	11	Died of disease (20 months)	None	Yes	No complications	RT	None	RT
**R0010** **(M, 53)**	Metastatic melanoma to the left pelvis	No	Yes	18	Died of disease (4 months)	None	Yes	Intraoperative venous bleeding. Emergent angiogram IR embolization 1st re-admission to hospital for concern of active bleeding in the surgical area. Treatment embolization of left illiac artery. 2nd re-admission to hospital for continued surgical incision drainage observation. New bleed ruled out. Treatment with silvadene dressing changes.	IMT	None	None
**R0012** **(M, 41)**	Metastatic Pleomorphic Spindle cell sarcoma	Yes	Yes	1	Alive (23 months) FDRS	None	Yes	No complications	CRT	None	RT, CT
**R0013** **(F, 55)**	Squamous cell Carcinoma (SCC) (History of Kidney Transplant)	Yes	Yes	2	Alive with disease (23 months) Recurrence of disease (1 month) in the hand and axillary lymph node. Receiving anti-rejection immunosuppression therapy for kidney transplant.	None	Yes	No complications	IMT	None	CT
**R0014** **(F, 77)**	Metastatic angiosarcoma to contralateral breast right Axilla	Yes	Yes	1	Alive (23 months) FDRS	None	Yes	No complications	RT, HT, CT	None	CT
**R0016** **(F, 50)**	Metastatic Non-Small Cell Lung Carcinoma (NSCLC) to Bone	No	Yes	2	Alive (22 months) FDRS	None	Yes	No complications	Internal RT (radioactive iodine); RT, CRT, and CT/IMT for NSCLC.	None	RT/IMT
**R0017** **(M, 26)**	Metastatic Desmoplastic Small Round Cell Sarcoma of left inguinal soft tissue and testicle	Yes	Yes	1	Alive (22 months) FDRS	None	Yes	No complications	CT	None	CRT
**R0018** **(M, 69)**	Metastatic Adenoid Cystic Carcinoma of the left submandibular gland	Yes	Yes	28	Alive (21 months) FDRS	None	Yes	Salivary leak. Return to the OR for Repair of floor of the mouth defect with buccal fat flap	RT, CT	None	None

R0 resection—No gross or microscopic tumor at the surgical margin; ECOG—Eastern Cooperative Oncology Group (Performance Status); LOS—Length of Stay; OS—Overall Survival after surgical resection; CHCP—Canady Helios Cold Plasma; HIPEC—Hyperthermic Intraperitoneal Chemotherapy; IMT—Immunotherapy; HT—Hormone Therapy; RT—Radiation Therapy; CT—Chemotherapy; IORT—Intraoperative Radiation Therapy; FDRS—Free of Disease at Resection Site; Region of Enrolment: S00## SMC, R00## RUMC.

## Data Availability

The data presented in this study are available in this article (and Appendix A).

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
