# Peer review of "The First Cold Atmospheric Plasma Phase I Clinical Trial for the Treatment of Advanced Solid Tumors: A Novel Treatment Arm for Cancer"

_cancers, 2023, doi:10.3390/cancers15143688_

Round 1
Reviewer 1 Report
Cold plasma used for cancer treatment has been studied for around 20 years, but its clinical trial has rarely been reported. From this point, this paper made a big progress which should be an important reference to the researchers in the field of plasma medicine. The experimental design is systematic, the therapeutic effect is good and the analysis is reasonable. Several questions or suggests are listed below:
1) It is better to provide the photograph of the plasma device and its physical/structural features.
2) How deep the anti-cancer effects of plasma can be? This is important because the residual tumor cells still might be protected by a certain thickness of tissue.
3) How to make sure that all incision tissue is treated in clinical operation? Because the diameter of the plasma jet is small, does it take a long time to “scan” the whole incision tissue?
Author Response
Response to Reviewer #1:
Thank you for your insightful comments and positive feedback on our paper regarding the use of CHCP for cancer treatment. We appreciate your time and effort in reviewing our work. We have carefully considered each of your questions and suggestions, and we respectfully address them below:
1) It is better to provide the photograph of the plasma device and its physical/structural features.
Response: Providing a photograph of the plasma device along with its physical and structural features is indeed a valuable suggestion. We understand the importance of visualizing the apparatus used in the experimental setup. In the revised version of the paper, we have included a detailed picture of the plasma device, accompanied by relevant descriptions highlighting its physical and structural characteristics in the revised Figure 1 of the main manuscript.
2) How deep the anti-cancer effects of plasma can be? This is important because the residual tumor cells still might be protected by a certain thickness of tissue.
Response: The depth of the anti-cancer effects of cold plasma is a crucial aspect to consider. We agree that residual tumor cells might potentially be protected by surrounding tissue. In our study, we focused primarily on assessing the therapeutic effects of CHCP treatment on the immediate vicinity of the CHCP treated site. However, we acknowledge the significance of investigating the penetration depth of CHCP treatment and its impact on residual tumor cells. We expanded this aspect in the revised manuscript, providing a range in the depth to which the anti-cancer effects of CHCP can reach. We measured the depth of tumor cell death in the ex vivo CHCP treated tumor tissue and found the depth of anti-cancer effect of CHCP could reach up to ~9.6mm. The details of the measurements and the figure are included in the updated supplemental data (Figure S8).
3) How to make sure that all incision tissue is treated in clinical operation? Because the diameter of the plasma jet is small, does it take a long time to “scan” the whole incision tissue?
Response: Ensuring the comprehensive treatment of all incision tissue during clinical operations is indeed a critical concern. Thank you for bringing up this important point, and we appreciate your understanding as we further explore and refine the practicality of implementing cold plasma treatment for comprehensive incision tissue coverage. We understand your query regarding the time required to "scan" the entire surgical margin of the tumor bed (Zone 0), considering the relatively small diameter of the plasma jet. In our clinical study, we demonstrated the efficacy of CHCP treatment on the immediate region surrounding the tumor bed (Zone 0). Additionally, it is important to note that the actual tumor bed region usually measures approximately 2 to 5 cm2 compared to the entire surgical site. Our experience dictates that the time needed for the treatment is usually 5 to 7 mins based on the cancer type. To date we have dosimetric data on 32 different cancer types. By applying the cold plasma uniformly to the entire tumor bed, we found that the treatment seamlessly integrates into the surgical procedure without significantly prolonging the entire surgical procedure. Moreover, our observations indicate that this standardized procedure has been highly effective, as we did not observe any local regional recurrences (LRR) in the CHCP-treated R0-MPM patients. Nonetheless, we acknowledge the need to address the practicality of treating the entire zone 0 area within a reasonable timeframe.
Reviewer 2 Report
1. In Figure 1. It is helpful to reader if treatment strategies of each therapy describe more precisely in Figure 1 or caption of Figure 1.
2. Authors argued that Canady Helios Cold Plasma (CHCP) is a novel CAP device. Would you introduce Canady Helios Cold Plasma (CHCP) as a separate Figure even though Canady Helios Cold Plasma already introduced in the previous article ? What is difference compared to other kind of cold plasma equipment ?
Author Response
Thank you for your insightful comments and positive feedback on our paper regarding the use of CHCP for cancer treatment. We appreciate your time and effort in reviewing our work. We have carefully considered each of your questions and suggestions, and we respectfully address them below:
- In Figure 1. It is helpful to reader if treatment strategies of each therapy describe more precisely in Figure 1 or caption of Figure 1.
Response: We appreciate the importance of providing more precise descriptions of the treatment strategies for each therapy in Figure 1. We apologize for any confusion caused by the lack of detailed information in the figure or its caption. In the revised version of the paper, we have taken steps to enhance the figure and its accompanying caption, aiming to provide a more explicit and comprehensive explanation of the treatment strategies employed for each therapy. This improvement will greatly aid readers in better understanding the specific approaches used in our study. Furthermore, we would like to provide additional context regarding the patients involved in the study. The patients included in our trial had undergone multiple rounds of chemotherapy and radiation prior to being considered for the CHCP treatment. The approval from the FDA was specifically granted for the use of CHCP at the surgical margins after the macroscopic removal of solid tumors in patients with Stage 4 metastatic or recurrent solid tumors. These patients were selected based on a comprehensive evaluation conducted by a multi-disciplinary team at each institution. They had a prior treatment history including chemotherapy, radiation, immuno-therapy, surgery, and Hyperthermic Intraperitoneal Chemotherapy (HIPEC), which made them eligible for participation in the CHCP trial. It is worth noting that traditionally, these patients would not have been offered surgery due to the advanced stage of their disease. The primary objective of the CHCP trial was to demonstrate the safety of the treatment. Additionally, if a complete (R0) or microscopic residual (R1) surgical resection could be achieved, it was hypothesized that the use of CHCP treatment might decrease the likelihood of local regional recurrence (LRR). We hope this additional information provides a clearer understanding of the patient selection and objectives of the study. Thank you for bringing these points to our attention, and we appreciate your guidance in improving the clarity and comprehensibility of our research.
- Authors argued that Canady Helios Cold Plasma (CHCP) is a novel CAP device. Would you introduce Canady Helios Cold Plasma (CHCP) as a separate Figure even though Canady Helios Cold Plasma already introduced in the previous article ? What is difference compared to other kind of cold plasma equipment ?
Response: We acknowledge your comment regarding the introduction of Canady Helios Cold Plasma (CHCP) as a novel CAP device and its potential inclusion as a separate figure. While it is true that Canady Helios Cold Plasma has been previously introduced in another article, we recognize the importance of highlighting its unique features and distinguishing it from other types of cold plasma equipment. In response to your suggestion, we have introduced Canady Helios Cold Plasma (CHCP) as a figure (Figure 1A) in the revised manuscript. This additional figure will explicitly showcase the distinctive characteristics and design elements that differentiate CHCP from other cold plasma equipment, providing readers with a clear visual representation of its novelty. Moreover, the temperature range maintained during treatment is between 26 to 30°C, the applicability of CHCP for various surgical approaches (open, laparoscopic, endoscopic, and minimally invasive), and its effectiveness in local regional recurrence (LRR) prevention. Furthermore, CHCP emphasizes the tumor type-specific treatment dosage, which is determined by the power and time provided through the software graphical user interface. This dosimetric approach ensures a standardized and controlled application of CHCP treatment.
We genuinely appreciate your insightful questions, as they have helped us identify areas for improvement and enhance the overall clarity of our work.